# Learning the specific quality of taste reinforcement in larval Drosophila

**Michael Schleyer[1]\*, Daisuke Miura[2], Teiichi Tanimura[2], Bertram Gerber[1,3,4]\***

[1]Department of Genetics of Learning and Memory, Leibniz Institute for Neurobiology, Magdeburg, Germany; [2]Department of Biology, Kyushu University, Fukuoka, Japan; [3]Institute for Biology and Behavior Genetics, Otto von Guericke University Magdeburg, Magdeburg, Germany; [4]Center of Behavioral Brain Sciences, Magdeburg, Germany

**Abstract** The only property of reinforcement insects are commonly thought to learn about is its value. We show that larval *Drosophila* not only remember the value of reinforcement (How much?), but also its quality (What?). This is demonstrated both within the appetitive domain by using sugar vs amino acid as different reward qualities, and within the aversive domain by using bitter vs high-concentration salt as different qualities of punishment. From the available literature, such nuanced memories for the quality of reinforcement are unexpected and pose a challenge to present models of how insect memory is organized. Given that animals as simple as larval *Drosophila*, endowed with but 10,000 neurons, operate with both reinforcement value and quality, we suggest that both are fundamental aspects of mnemonic processing—in any brain.

## Introduction

What are the fundamental capacities of insect brains? To date, little use has been made of insect memory experiments to reveal these capacities, in particular regarding reinforcement. For example, after experiencing an odor with a sugar reward, fruit flies (*Drosophila melanogaster*) approach that odor in a later test. All the known circuitry (*Heisenberg, 2003*; *Perisse et al., 2013*) of such learned search behavior suggests that this is because the odor has acquired positive value; that is, that the flies are expecting to find 'something good' in its vicinity (*Heisenberg, 2003*; *Gerber and Hendel, 2006*; *Schleyer et al., 2011*; *Perisse et al., 2013*). Likewise, *Drosophila* can associate an odor with an electric-shock punishment. This supports their learned escape in a later test because they may expect 'something bad' with the odor. In other words, the only feature of reinforcement processing that insects are granted is value. We show that larval *Drosophila* are in a defined sense richer than this in their mnemonic capacity: they also recall of what particular quality that good or bad experience was. Given the numerical simplicity of the larval brain, this is suggested to be a more basic property of brains than hitherto assumed.

To address this question, we exploit an established assay for Pavlovian conditioning (*Gerber et al., 2013*) that allows reinforcers of various strength and quality to be used. In this Petri dish assay, larvae are placed onto a tasteless agarose substrate. This substrate is supplemented with a fructose sugar reward—if odorant A is presented. Odorant B is presented without the reward (A+/B). For a companion group of larvae, contingencies are reversed (A/B+). In a binary choice test, the larvae then systematically approach the previously rewarded odorant (*Figure 1—figure supplement 1*). This behavior, quantified as a positive associative performance index (PI) (*Figure 1A*), can best be grasped as a memory-based search for reward: if the test is performed in the presence of fructose, the learned approach is abolished (*Figure 1A*) (*Gerber and Hendel, 2006*; *Schleyer et al., 2011*) (olfactory behavior per se is not affected: see below). This is adaptive as learned search behavior is indeed obsolete

**\*For correspondence:** michael. schleyer@lin-magdeburg.de (MS); bertram.gerber@lin-magdeburg.de (BG)

**Competing interests:** The authors declare that no competing interests exist.

**Reviewing editor**: Alexander Borst, Max Planck Institute of Neurobiology, Germany

**eLife digest** Actions have consequences; positive consequences or rewards make it more likely that a behavior will be repeated, while negative consequences or punishments can stop a behavior occurring again. Neuroscientists commonly refer to such rewards and punishments as 'reinforcement'.

Fruit flies that are given a reward of sugar when they experience an odor will move towards the odor in later tests. However, in 2011, research revealed that if the flies were given at least the same amount of sugar in the tests as they were rewarded with during the earlier training, the flies stopped moving towards the odor. This suggests that fruit flies can recall how strong a reward was in the past and compare this remembered strength to the current reward on offer; fruit flies will only continue searching if they expect to gain a larger reward by doing so.

Insects were commonly thought to only learn the amount or 'value' of reinforcement, but not recall what kind or 'quality' of reward (or punishment) they had experienced. Now Schleyer et al.—including some of the researchers involved in the 2011 work—challenge and extend this notion and show that fruit fly larvae can remember both the value and quality of rewards and punishments.

Fruit fly larvae were trained to expect a reward of sugar when exposed to one odor and nothing when exposed to a different odor. Consistent with the previous results, the larvae moved towards the first odor in the tests where no additional reward was provided. Moreover, the larvae did not move towards the odor in later tests if an equal or greater amount of sugar was provided during the testing stage.

Schleyer et al. then took larvae that had been trained to expect a sugar reward and gave them a different, but equally valuable, reward during the testing stage—in this case, the reward was an amino acid called aspartic acid. These experiments revealed that most of the larvae continued to move towards the sugar-associated odor in search of the sugar reward. This indicates that the larvae were able to remember the quality of the reward, namely that it was sugar rather than aspartic acid.

Schleyer et al. performed similar experiments, and observed similar results, when using two different punishments: bitter-tasting quinine and high concentrations of salt. These findings show that experiencing an odor along with taste reinforcement could set up a memory specific to the quality of reinforcement in fruit fly larvae. Given the numerical simplicity of a larva's brain—which contains only 10,000 neurons—it is likely that other animals can also recall both the value and quality of a reward or punishment. However, understanding how such specificity comes about should be easier in the larva's simple brain.

in the presence of a sought-for item. We have previously shown (*Schleyer et al., 2011*) that regardless of the absolute concentration of fructose, such an abolishment is seen if the fructose concentration in the test substrate is equal to or higher than that used in training. This means that learned behavior is based on a relative assessment: the larvae recall how strong the training reward was and compare this remembered strength to the current testing situation. Only if that comparison promises a gain (remembered strength > current strength) do they search for what they can thus expect to gain at the odor source. We note that widely applied formal learning models of the Rescorla-Wagner type (*Rescorla and Wagner, 1972*) propose that memory acquisition will only occur if something new and unexpected happens, specifically if the experienced reward is stronger than predicted on the basis of memory (current strength > remembered strength). Thus, the same two pieces of information are compared during memory acquisition on the one hand and the expression of learned search behavior on the other hand—yet in a 'swapped' way. This can inform the animals respectively about what is new or what there is to be gained. Here, we ask whether these processes are integrated across different qualities of reward into one common scale of appetitive value or whether separate systems exist to confer mnemonic specificity for the 'quality' of reward.

## Results

We introduce aspartic acid, a proteinogenic amino acid, as a novel quality of reward (*Figure 1—figure supplement 2*). Concentrations of aspartic acid and fructose are chosen such that their reward value

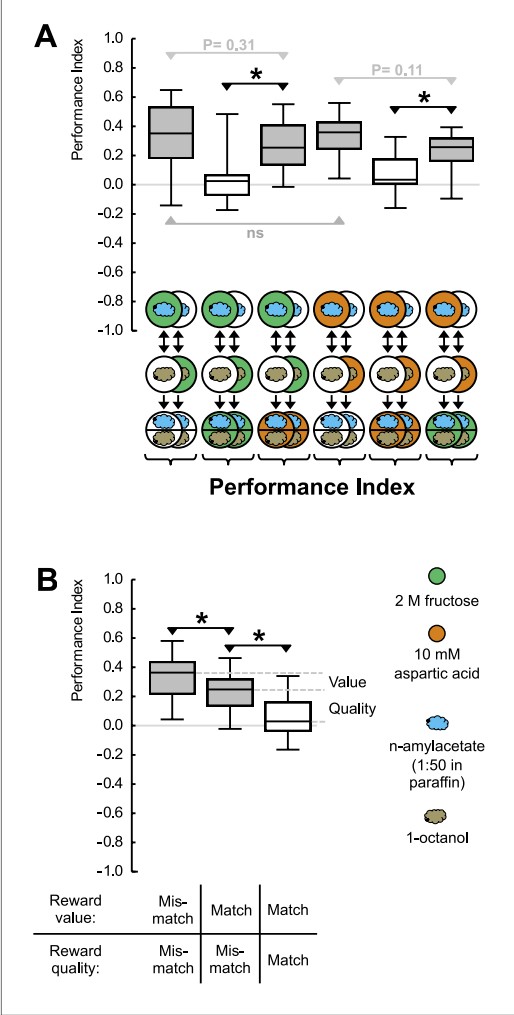

**Figure 1**. Reward processing by quality and value. (**A**) Larvae are trained to associate one of two odors with either 2 M fructose or 10 mM aspartic acid as reward. Subsequently, they are tested for their choice between the two odors—in the absence or in the presence of either substrate. For example, in the left-most panel, a group of larvae is first (upper row) exposed to n-amyl acetate (blue cloud) together with fructose (green circle), and subsequently (middle row) to 1-octanol (gold cloud) without any tastant (white circle). After three cycles of such training, larvae are given the choice between n-amyl acetate and 1-octanol in the absence of any tastant (lower row). A second group of larvae is trained reciprocally, that is, 1-octanol is paired with fructose (second column from left, partially hidden). For the other panels, procedures are analogous. Aspartic acid is indicated by brown circles. (**B**) Data from (**A**) plotted combined for the groups tested on pure agarose ('Mismatch' in both value and quality), in the presence of the respectively other reward, or of the training reward. Learned search behavior towards the reward-associated odor is abolished in presence of the training reward because

*Figure 1. Continued on next page*

is equal (*Figure 1A*). This allows us to study the larvae under test conditions that are of equal reward value but that either match or do not match the 'quality' of reward that has been employed during training. If the larvae merely searched for a reward of remembered value, learned search behavior should cease in both cases—because current strength matches remembered strength in both cases. If reward quality were the sole determinant for learned search, in contrast, learned search should be abolished only when test and training substrate match in quality, but should remain intact when quality does not match. If the larvae searched for a reward specified both by its value and by its quality, scores in the mismatch case should be partially abolished: in that case, the reward's value is as sought yet its quality is not. We find that learned search is fully abolished when the training and test reward match in both value and quality but remains partially intact (by 68%) if there is a mismatch in reward quality between training and test (*Figure 1A,B*). We conclude that after odor-fructose training the larvae approach the odor both in search of something 'good' (value) and in search of what is specifically fructose (quality of reward). Likewise, after odor-aspartic acid training, they search both for something 'good' and for aspartic acid. In other words, if during the test the larvae, for example, have sugar anyway but remember where aspartic acid can be found, they will still go for aspartic acid in addition.

Regarding the aversive domain, pairing an odor with quinine as punishment leads to aversive memory (*Gerber and Hendel, 2006*; *Schleyer et al., 2011*; *El-Keredy et al., 2012*; *Apostolopoulou et al., 2014*). In this case, learned behavior can best be understood as an informed escape that is warranted in the presence but not in the absence of quinine. Accordingly, in the presence but not in the absence of quinine one observes that the larvae run away from the previously punished odor (*Gerber and Hendel, 2006*; *Schleyer et al., 2011*; *El-Keredy et al., 2012*; *Apostolopoulou et al., 2014*) (see also *Niewalda et al., 2008*; *Schnaitmann et al., 2010*; *Eschbach et al., 2011*; *Russell et al., 2011* for reports using other aversive reinforcers and/or adult flies). Notably and in accordance with our earlier results in the appetitive domain (*Schleyer et al., 2011*), such learned escape does not merely depend on the concentration of quinine in the test; rather, learned escape lessens as the quinine concentration in the test is reduced *relative* to that in training (*Figure 2—figure supplement 1*).

*Figure 1. Continued*

both reward value and reward quality in the testing situation are as sought-for ('Match' in both cases), yet search remains partially intact in the presence of the other quality of reward, because reward value is as sought-for ('Match') but reward quality is not ('Mismatch'). Please note that value-memory is apparently weaker than memory for reward quality, and is revealed only when pooling across tastants. Sample sizes 15–19. Shaded boxes indicate p < 0.05/6 (**A**) or p < 0.05/3 (**B**) from chance (one-sample sign-tests), asterisks indicate pairwise differences between groups at p < 0.05/3 (**A**) or p < 0.05/2 (**B**) (Mann–Whitney U-tests).
The following figure supplements are available for figure 1:

**Figure supplement 1**. Preference scores for the reciprocally trained groups of **Figure 1**.

**Figure supplement 2**. (**A**, **B**) Larvae are trained in a one-odor version of the learning paradigm, using different concentrations of aspartic acid as reward.

Given that high-concentration salt can also serve as punishment (*Gerber and Hendel, 2006*; *Niewalda et al., 2008*; *Russell et al., 2011*), we ask whether an odor-quinine memory is specific in prompting escape from quinine—but not from salt. For concentrations of quinine and salt that are of equal value as punishment, this is indeed the case (*Figure 2A*) (*Figure 2—figure Supplement 2*); likewise, odor-salt memories are specific in prompting learned escape from salt but not from quinine (for a summary see *Figure 2B*). Such specificity shows that larvae have a memory specific to the quality of punishment, a memory that can specifically be applied in the appropriate situation. We stress that the present results do not provide proof of the absence of aversive 'common currency' value processing. Indeed, in cases of unequal punishment value, larvae may use this information (*Eschbach et al., 2011*).

Taken together, within both the appetitive and aversive domain, experiencing an odor with a taste reinforcement can establish an associative olfactory memory that is specific to the quality (fructose, aspartic acid, quinine, high-concentration salt) of taste reinforcement.

The experimental twist to reveal such quality-of-reinforcement memory is accomplished by flagrantly breaking the first rule of associative memory research: namely never, ever, to test for learned behavior in the presence of the reinforcer. We would like to stress that innate olfactory behavior per se is not affected by the presence of any of the tastant reinforcers (*Hendel et al., 2005*; *Schleyer et al., 2011*) (*Figure 3*). Also, the mere presence of any given tastant reinforcer during the memory test is not a critical determinant for whether learned behavior is observed: learned behavior can be observed (or not) in the presence of any of the tastant reinforcers in this study—what matters is how closely it matches the one used during training in quality and/or in value (*Figures 1B and 2B*).

## Discussion

The mushroom bodies, a third-order 'cortical' (*Tomer et al., 2010*) brain region in insects, are canonically proposed to feature distinct regions harboring appetitive and aversive olfactory memory traces, respectively (*Heisenberg, 2003*; *Perisse et al., 2013*; see also *Schleyer et al., 2011*) (*Figure 4A,B*). Only recently has the possibility of different neuronal substrates underlying different qualities of reinforcement come to be considered. These studies have so far not yielded a double dissociation between different dopaminergic mushroom body input neurons for different qualities of reinforcement:

- For the aversive domain *Galili et al. (2014)* suggested that the set of dopaminergic mushroom body input neurons responsible for heat-punishment in adult *Drosophila* is nested within that for electric-shock punishment. Similarly, electric-shock punishment and punishment with the insect repellent DEET appear to be signaled towards the mushroom body by largely if not completely overlapping sets of dopamine neurons (*Das et al., 2014*).
- For the appetitive domain, a set of dopaminergic mushroom body input neurons (included in the 0104-Gal4 strain) that was previously found to be required for sugar-learning in adult *Drosophila* (*Burke et al., 2012*) turned out to be dispensable for water-reward learning (*Lin et al., 2014*). Whether in turn dopaminergic mushroom body input neurons included in the R48B04 strain, which were discovered by *Lin et al. (2014)* to be required for water-reward learning, are dispensable for sugar-learning remains to be tested.

Thus, the nuanced memory of at least two qualities of appetitive and two qualities of aversive taste reinforcers as shown in the present study is unexpected. Appropriate to such nuanced memories, the

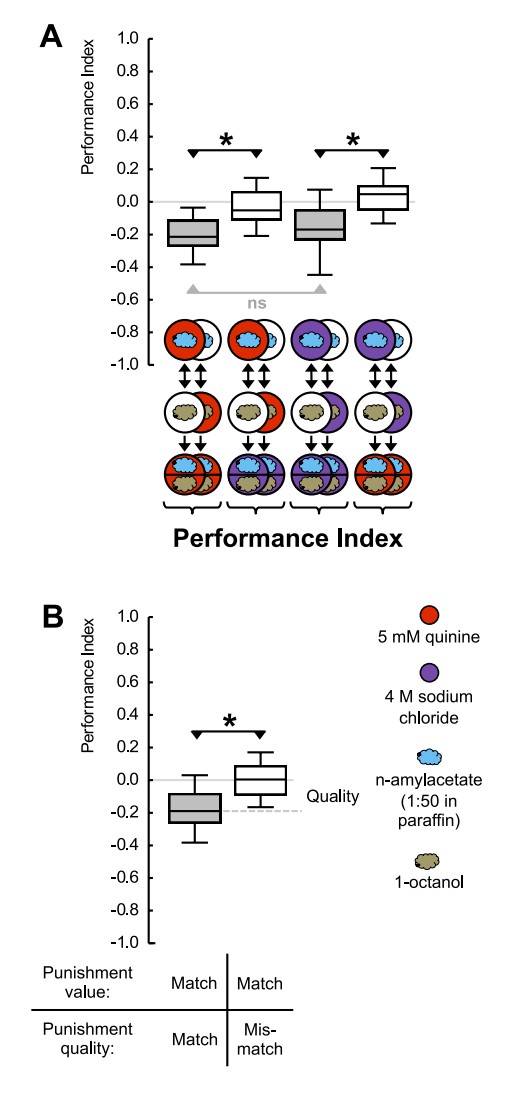

**Figure 2**. Punishment processing by quality. (**A**) Larvae are trained to associate one of two odors with either 5 mM quinine (red circle) or 4 M sodium chloride (purple circle) as punishment and asked for their choice between the two odors—in the presence of either substrate. The larvae show learned escape from the punishment-associated odor only if a matching quality of punishment is present during the test as compared to training. (**B**) Data from (**A**) combined according to 'Match' or 'Mismatch' between test- and training-punishment. We note that value-memory would reveal itself by negative scores upon a match of punishment value despite a mismatch in punishment quality, which is not observed (right-hand box plot, based on second and fourth box plot from **A**). Sample sizes: 25–32. Shaded boxes indicate p < 0.05/4 (**A**) or p < 0.05/2 (**B**) from chance (one-sample sign-tests), asterisks indicate pairwise differences between groups at p < 0.05/3 (**A**) or p < 0.05 (**B**) (Mann–Whitney U-tests). For a detailed description of the sketches, see legend of *Figure 1*.
*Figure 2. Continued on next page*

mushroom bodies show a fairly complex substructure, even in larval *Drosophila*. At least 10 mushroom body regions are recognized, defined by the tiled innervation of input and output neurons (*Pauls et al., 2010b*). Our behavioral data suggest that at least five such tiles of the mushroom body would be required to accommodate learned search for fructose or aspartic acid, learned escape from quinine or from high-concentration salt and in addition a less specific appetitive value-memory (*Figure 4C*). Clearly the things worth remembering for a larva include many more than these five (*Niewalda et al., 2008*; *Pauls et al., 2010a*; *Eschbach et al., 2011*; *Khurana et al., 2012*; *Rohwedder et al., 2012*; *Diegelmann et al., 2013*). Likewise, the behavioral repertoire of larvae may be considerably greater than thought (*Vogelstein et al., 2014*). Using our current approach, it will now be possible to systematically determine the limits of specificity in the processing of reinforcement quality. This may reveal signals of intermediate specificity to inform the animals about, for example, edibility, caloric value, proteinogenic value, suitability for pupariation, toxicity, or even acutely and situationally modulated matters of concern (*Simpson et al., 2015*).

We note that the distinction between fructose and aspartic acid memory implies that the sensory neurons mediating the rewarding effects of these stimuli cannot be completely overlapping and that the sensory neurons mediating the punishing effects of quinine and high-concentration salt likewise cannot (for reviews of the taste system in *Drosophila*, see *Cobb et al., 2009*; *Gerber et al., 2009*). For identifying these neurons, it is significant that they may be distinct from those mediating innate choice behavior (*Apostolopoulou et al., 2014*; *König et al., 2014*).

In the vertebrate literature, the processing of reward by value has been regarded as a matter of sophistication because an integrated, higher-order value signal can be generated from sensorially distinct qualities of reward (e.g., *Lak et al., 2014*). On the other hand, reward expectations can apparently also be processed in a quality-specific manner (e.g., *Dickinson and Balleine, 1994*; *Watanabe, 1996*). In terms of the minimally required number of cells, the processing by reinforcer quality is more demanding than value-only processing (*Figure 4B,C*). The fact that even the humble, 10,000-neuron brain of a larva operates with both reward value and quality may suggest that they both represent fundamentally important, indispensable aspects of reward processing.

*Figure 2. Continued*

The following figure supplements are available for figure 2:

**Figure supplement 1**. Quinine memory includes quinine strength.

**Figure supplement 2**. Preference scores for the reciprocally trained groups of *Figure 2*.

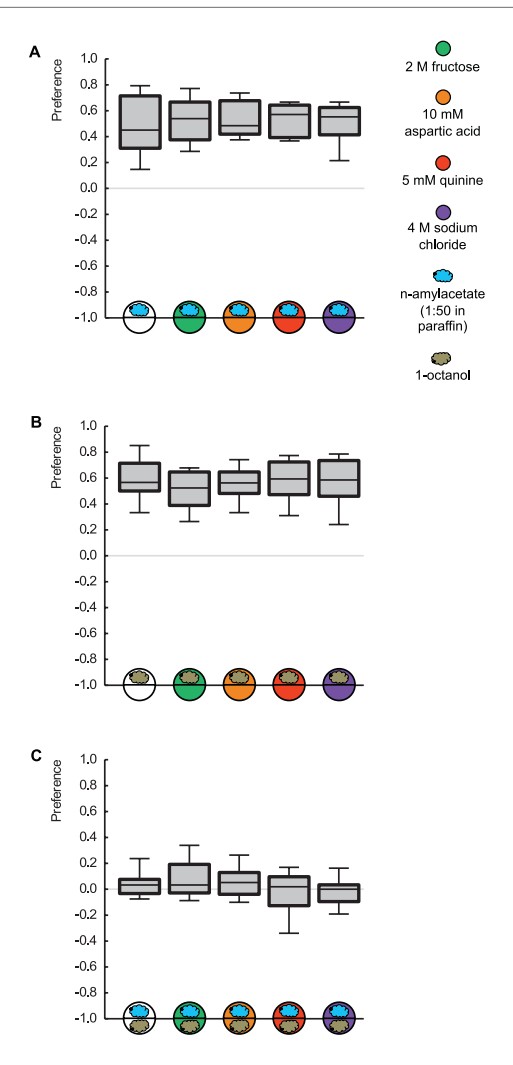

**Figure 3**. Innate odor preference is not influenced by taste processing. Larvae are tested for their olfactory preference regarding (**A**) n-amyl acetate (blue cloud), (**B**) 1-octanol (gold cloud), or (**C**) for their choice between n-amyl acetate and 1-octanol. This is done in the presence of pure agarose (white circle), 2 M fructose (green circle), 10 mM aspartic acid (brown circle), 5 mM quinine (red circle), or 4 M sodium chloride (purple circle). We find no differences in odor preferences across different substrates ($p > 0.05$, Kruskal–Wallis tests). Sample sizes: 20–26.

# Materials and methods

## General

We used third-instar feeding-stage larvae from the Canton-Special wild-type strain, aged 5 days after egg laying. Flies were maintained on standard medium, in mass culture at 25°C, 60–70% relative humidity and a 12/12 hr light/dark cycle. Before each experiment, we removed a spoonful of food medium from a food vial, collected the desired number of larvae, briefly rinsed them in distilled water, and started the experiment.

For experiments, we used Petri dishes of 90-mm inner diameter (Sarstedt, Nümbrecht, Germany) filled with 1% agarose (electrophoresis grade; Roth, Karlsruhe, Germany). As reinforcers fructose (FRU; CAS: 57-48-7; Roth, Karlsruhe, Germany), aspartic acid (ASP; CAS: 56-84-8; Sigma–Aldrich, Seelze, Germany), quinine (QUI; CAS: 6119-70-6; Sigma–Aldrich), or sodium chloride (SAL; 7647-14-5; Roth, Karlsruhe, Germany) were used in concentrations given in the results section. As odors, we used *n*-amyl acetate (AM; CAS: 628-63-7; Merck, Darmstadt, Germany), diluted 1:50 in paraffin oil (Merck, Darmstadt, Germany) and 1-octanol (OCT; CAS: 111-87-5; Sigma–Aldrich).

## Learning

Prior to experiments, odor containers were prepared: 10 µl of odor substance was filled into custom-made Teflon containers (5-mm inner diameter with a lid perforated with seven 0.5-mm diameter holes). Before the experiment started, Petri dishes were covered with modified lids perforated in the center by 15 holes of 1-mm diameter to improve aeration.

For training, 30 larvae were placed in the middle of a FRU-containing dish with two odor containers on opposite sides, both filled with AM. After 5 min, larvae were displaced onto an agarose-only dish with two containers filled with OCT, where they also spent 5 min. Three such AM+/OCT training cycles were performed, in each case using fresh dishes. In repetitions of the experiment, in half of the cases training started with a reinforcer-added dish (AM+/OCT) and in the other half with an agarose-only dish (OCT/AM+). For each group of larvae trained AM+/OCT (or OCT/AM+, respectively), a second group was trained reciprocally, that is, AM/OCT+ (or OCT+/AM, respectively).

Following training, larvae were transferred to a test Petri dish that, as specified for each experiment, did or did not contain a reinforcer and given the choice between the two trained odors.

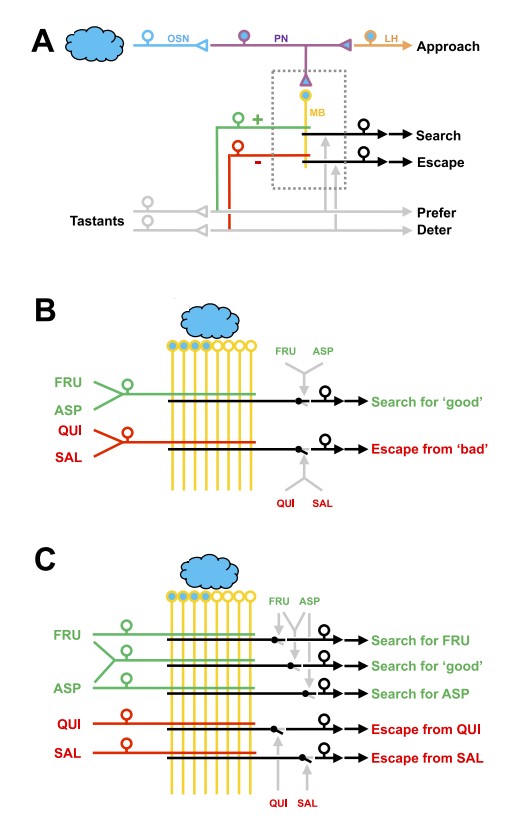

**Figure 4**. Working hypotheses of reinforcement processing by value-only or by value and quality in larval *Drosophila*. (**A**) Simplified overview (based on e.g., **Heisenberg, 2003**; **Perisse et al., 2013**). Odors are coded combinatorially across the olfactory sensory neurons (OSN, blue). In the antennal lobe, these sensory neurons signal towards local interneurons (not shown) and projection neurons (PN, deep blue). Projection neurons have two target areas, the lateral horn (LH, orange) mediating innate approach, and the mushroom body (MB, yellow). Reinforcement signals (green and red for appetitive and aversive reinforcement, respectively) from the gustatory system reach the mushroom body, leading to associative memory traces in simultaneously activated mushroom body neurons. In the present analysis, this sketch focuses selectively on five broad classes of chemosensory behavior, namely innate odor approach, learned odor search and escape, as well as appetitive and aversive innate gustatory behavior. The boxed region is displayed in detail in (**B–C**). The break in the connection between mushroom body output and behavior is intended to acknowledge that mushroom body output is probably not in itself sufficient as a (pre-) motor signal but rather exerts a modulatory effect on weighting between behavioral options (**Schleyer et al., 2013**; **Menzel, 2014**; **Aso et al., 2014**). (**B**) Reinforcement processing by value (based on e.g., **Heisenberg, 2003**; **Schleyer et al., 2011**; **Perisse et al., 2013**): a reward neuron sums input from fructose and aspartic acid pathways and thus

*Figure 4. Continued on next page*

After 3 min, larvae were counted and a preference score calculated as:

(1) Preference = $(\#_{AM} - \#_{OCT})/\#_{Total}$.

In this equation, # indicates the number of larvae on the respective half of the dish. Thus, PREF values are constrained between 1 and −1 with positive values indicating a preference for AM and negative values indicating a preference for OCT.

From two reciprocally trained groups of animals, we calculated an associative performance index (PI) as:

(2) Performance Index = $(\text{Preference}_{AM+/OCT} - \text{Preference}_{AM/OCT+})/2$.

Thus, performance index values can range from 1 to −1 with positive values indicating appetitive and negative values indicating aversive conditioned behavior.

Preferences and performance indices for other reinforcers were calculated in an analogous way.

## Innate odor preference

A group of 30 experimentally naïve larvae were placed on a Petri dish filled with pure agarose (PUR) or agarose containing FRU, ASP, QUI, or SAL. Animals were given the choice between an odor-filled and an empty Teflon container; as odor, either AM or OCT was used. After 3 min, the position of the larvae was scored to calculate their preference as:

(3) Preference = $(\#_{AM} - \#_{EM})/\#_{total}$ (this equation was used in *Figure 3A*).

(4) Preference = $(\#_{OCT} - \#_{EM})/\#_{total}$ (this equation was used in *Figure 3B*).

To measure choice, one container was loaded with AM and the other with OCT and preference calculated as:

(5) Preference = $(\#_{AM} - \#_{OCT})/\#_{total}$ (this equation was used in *Figure 3C*).

## Statistical analyses

Preference values and performance indices were compared across multiple groups with Kruskal–Wallis tests. For subsequent pair-wise comparisons, Mann–Whitney U-tests were used. To test whether values of a given group differ from zero, we used one-sample sign tests. When multiple tests of the same kind are performed within one experiment, we adjusted significance levels by a Bonferroni correction to keep the experiment-wide error rate below 5%. This was done by dividing the critical p value 0.05 by the number of tests. We present our data as box plots which represent the median as the middle line and 25%/75% and 10%/90% as box boundaries and whiskers, respectively.

*Figure 4. Continued*

establishes a memory allowing for learned search for 'good'. In a functionally separate compartment, a punishment neuron summing quinine and salt signals likewise establishes a memory trace for learned escape from 'bad'. This scenario cannot account for quality-of-reinforcement memory. (**C**) Reinforcement processing by both value and quality: in addition to a common, value-specific appetitive memory, fructose and aspartic acid drive discrete reward signals leading to discrete memory traces in at least functionally distinct compartments of the Kenyon cells, which can be independently turned into learned search. For aversive memory, there may be only quality-specific punishment signals. This scenario is in accordance with the present data.

## Acknowledgements

This study received institutional support by the Otto von Guericke Universität Magdeburg, the Wissenschaftsgemeinschaft Gottfried Wilhelm Leibniz (WGL), the Leibniz Institut für Neurobiologie (LIN), the Kyushu University, as well as grant support from the Deutsche Forschungsgemeinschaft (DFG) (SFB 779 Motivated behavior), the Bundesministerium für Bildung und Forschung (BMBF, through the Bernstein Focus Program Insect-Inspired Robotics), the European Commission grant MINIMAL (FP7—618045) (to BG), and by a Grant-in-Aid for Scientific Research from the Ministry of Education, Culture, Sports, Science and Technology of Japan (to TT). Experimental contributions of M Döring, J Leibiger, and K Tschirner, as well as comments from R Glasgow and A Thum are gratefully acknowledged. Procedures comply with applicable law.

## Additional information

### Funding

| Funder | Grant reference number | Author |
| --- | --- | --- |
| Deutsche Forschungsgemeinschaft | SFB 779 | Michael Schleyer, Bertram Gerber |
| Bundesministerium für Bildung und Forschung | Bernstein Fokus Insect-inspired robotics | Michael Schleyer, Bertram Gerber |
| European Commission | MINIMAL FP7 - 618045 | Michael Schleyer, Bertram Gerber |
| Kyushu University | | Daisuke Miura, Teiichi Tanimura |

The funders had no role in study design, data collection and interpretation, or the decision to submit the work for publication.

### Author contributions

MS, Conception and design, Acquisition of data, Analysis and interpretation of data, Drafting or revising the article; DM, Acquisition of data, Analysis and interpretation of data, Drafting or revising the article; TT, BG, Conception and design, Analysis and interpretation of data, Drafting or revising the article

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
