## [Decision Letter]

Thank you for sending your work entitled “Type-of-reinforcement memory in larval *Drosophila*” for consideration at *eLife*. Your article has been favorably evaluated by a Senior editor, a Reviewing editor, and 3 reviewers.

In summary, the reviewers say: “This manuscript reports a well-designed study on the role of parameters of reinforcement stimuli that are learned by the larvae of *Drosophila*. The authors convincingly show that not only the quantity of a rewarding or punishing reinforcer is learned (reported by earlier of the same authors) but also the quality, more specifically two different qualities in the rewarding and the punishing stimuli. This is an important finding that opens up mechanistic studies on the coding of reinforcing stimuli in this very small nervous system of the larvae. The researchers' principal finding is both interesting and well-substantiated by their data.”

However, there were also a number of points raised by the reviewers that are summarized in the following. When you revise your manuscript, we would like you to address each of them and give a point-by-point response in a separate letter.

Concerning data evaluation, interpretation, and figures:

1) Why do the authors pool the data of Figure 1 according to whether there is a match or mismatch of the reward during training / test? The authors should perform the comparisons of components (i.e. 1st to 3rd bar and 4th to 6th as done for aversive reinforcers in Figure 2) and adjust their conclusions accordingly.

2) The authors claim (Results, end of paragraph 2) that “(...) learned escape ceases if the quinine concentration in the test is relatively smaller than during training.” However, in Figure 2—figure supplement 1, they show that training larvae with 5 mM quinine and testing them in the presence of 0.05 mM quinine still results in avoidance that is significantly different from 0 (4th bar). This result directly contradicts the authors' claim.

3) Figure 4: The three figures are all based on an assumption that should be mentioned, namely that the mushroom body produces labeled lines of (pre)motor commands. This assumption is in line with the tradition of “modelling” in Drosophila research but is not in agreement with other interpretations about the function of the mushroom body. The weakness of this kind of thinking degrades the interesting and important data reported here. Therefore, we suggest removing Figure 4. If, however, the authors want to keep it than they need to put it in full context about the concepts of mushroom body function (including other insect species) and into the context of what is known about the mushroom body extrinsic neurons (also including other insect species).

Concerning context and topics for discussion as well as future experiments:

1) The authors do not fully discuss similar findings in mammals. In primates, a potential neuronal correlate for reward-type specificity has been shown (e.g. Watanabe, Nature 1996). Given these findings the authors' views on larval type-of-reinforcement learning should be placed in a broader context.

2) Likewise the authors should comment on recent relevant papers in *Drosophila* about the distinct/common neuronal correlates for sugar/water (19), and electric shock/ DEET (5)/high temperature (10).

3) What is known about taste receptors in fruit flies, and specifically, which of them are expressed in fruit fly larvae? *C elegans* expresses lots of different receptors for stimuli of the same valence (good/bad) on the same neuron. Therefore, if one were to do the same experiments on *C elegans*, that they would not be capable of encoding type of reinforcement, only the value. This would be an important prediction from the authors work.

4) Fruit flies, like other insects, have a lot of different taste receptors. They presumably need them to discriminate among many different tastants. So, in some sense, one would predict a priori that insects can discriminate value. Why would the receptors that enabled the 'type' discrimination evolve only to have them summed in a 'good' and in a 'bad' center in the brain for association? It might suggest an interesting experiment for future work that reflects on the need to discriminate type. There is a large literature on nutrition of carbohydrate, amino acids, etc., in insects. A Google search for authors Simpson or Raubenheimer and 'nutritional rails' will find relevant papers. Insects need to maintain a balance of these nutrients in their diet. If one experimentally drives insects off of their preferred nutritional balance (ratio of carbohydrate to amino acid, for example), insects will typically change their diet choices to bring them back into nutritional balance. If the authors were to deprive larvae of sugar or amino acids in their diet, then one might expect interference where there is currently none (e.g. Figure 1 the 3rd and 6th experimental groups). Having shown that *Drosophila* larvae can encode type, which the authors conclusively did, then showing that insects can ignore the learned preferences for a deprived type would show even more complexity. But this is not an experiment that would be needed for a revision.

---

## [Author Response]

*Concerning data evaluation, interpretation and figures*:

*1) Why do the authors pool the data of*
Figure 1
*according to whether there is a match or mismatch of the reward during training / test? The authors should perform the comparisons of components (i.e. 1st to 3rd bar and 4th to 6th as done for aversive reinforcers in*
Figure 2*) and adjust their conclusions accordingly*.

We now display the respective P values for the two mentioned comparisons within Figure 1.

Please bear in mind that the default understanding, in the fly community, is that memory is about reinforcement value alone. The outcome of pooling across tastants in Figure 1 is thus conservative in the sense that it supports this default notion of a value‐memory. We have experienced that not-pooling the data despite the trends apparent in Figure 1 can come across as unfair to this default notion of value‐memory, which is as giving an iconoclastic spin to the conclusions. We hope that the way we now present the data in Figure 1 and the way we comment on the situation is acceptable.

Please allow two more technical comments:

We agree with the comment that the comparisons indicated in Figure 1 and in Figure 2 should be the same. Now the comparisons in Figure 2 are comparisons between the respective Match/Match case versus the Match/Mismatch case. That same comparison is the one indicated with an asterisk in Figure 1.

In Figure 2 the existence of a value‐memory would have revealed itself by the scores being significantly negative even if the quality of the test punishment were a mismatch with the training punishment; we have added a sentence to clarify this. In other words, for revealing value‐memory in the aversive case, the test in the absence of any reinforce (corresponding to the first bar in Figure 1) is not needed.

*2) The authors claim (Results, end of paragraph 2) that “(...) learned escape ceases if the quinine concentration in the test is relatively smaller than during training.” However, in*
Figure 2—figure supplement 1*, they show that training larvae with 5 mM quinine and testing them in the presence of 0.05 mM quinine still results in avoidance that is significantly different from 0 (4th bar). This result directly contradicts the authors' claim*.

We agree and regret that the sentence was poorly phrased. We intended to say that learned escape is lessened as the concentration of quinine during the test is reduced relative to that during training. The text has now been amended.

*3)*
Figure 4*: The three figures are all based on an assumption that should be mentioned, namely that the mushroom body produces labeled lines of (pre)motor commands. This assumption is in line with the tradition of “modelling” in Drosophila research but is not in agreement with other interpretations about the function of the mushroom body. The weakness of this kind of thinking degrades the interesting and important data reported here. Therefore, we suggest removing*
Figure 4*. If, however, the authors want to keep it than they need to put it in full context about the concepts of mushroom body function (including other insect species) and into the context of what is known about the mushroom body extrinsic neurons (also including other insect species)*.

We certainly agree that the mushroom body is richer in function than our sketch suggests, and we actually also agree that mushroom body output by itself is probably not a sufficient (pre)motor command. We opt to keep the figure. We have modified Figure 4, changed its title and rewritten the legend to explicate our view on what mushroom body output means to the animal.

To broaden the perspective, this includes additional review‐references that try to capture some of the literature beyond what is of most immediate relevance. Given the nature of the current paper as a short report, we wonder whether any more exhaustive discussion may be disproportionate. We hope this will be acceptable.

*Concerning context and topics for discussion as well as future experiments*:

*1) The authors do not fully discuss similar findings in mammals. In primates, a potential neuronal correlate for reward-type specificity has been shown (e.g. Watanabe, Nature 1996). Given these findings the authors' views on larval type-of-reinforcement learning should be placed in a broader context*.

We agree the account of the vertebrate literature was too selective. We have now tried to find a better balance between the needs to do justice to the status of neighboring fields of study (including balancing reference to Lak et al with the one by Watanabe, and making it explicit that both are merely examples), and to be brief in the context of the present short report.

*[Editors’ note: the following was sent as an optional point for the discussion, not an essential revision, after the original decision letter was sent*.*]*

*“US (or outcome) devaluation is a common experimental procedure to address the stimulus-specific learned outcome value (mainly using rodent and primate systems). So by definition, learned reinforcement in these studies is stimulus‐specific (reinforcement quality)*.

*Devaluation and its underlying neural substrate have been extensively studied in experimental psychology (e.g. Dickinson and Balleine Animal Learning & Behaviour 1994), but nowhere mentioned in this manuscript*.

*Therefore, I would like to see the discussion of the authors' findings in the context of all these vertebrate behavioural studies*.*”*

We agree that devaluation experiments would be a good approach to studying memory for reinforce quality. Indeed, we have tried to devalue a fructose reward by pairing it with quinine. However, this kind of experiment has failed to work in larvae so far (unpublished).

Meanwhile, we have found that bitter substances inhibit sugar detection ([17], Chemical Senses). Thus, the devaluation experiments would have to be repeated with a devaluing stimulus without such side effects. Anyway, our current approach circumvents this problem and has the advantage of also being applicable to punishments.

We have now added a reference to the devaluation literature in the context of our note regarding reward expectations specific for reward quality. We hesitate to engage in a more detailed discussion within the paper, as we feel it would be beyond the scope of a short report.

*2) Likewise the authors should comment on recent relevant papers in Drosophila about the distinct/common neuronal correlates for sugar/water (*[19]*), and electric shock/ DEET (*[5]*)/high temperature (*[10]*)*.

We have now included a fairly extensive discussion of these papers, which came out very shortly before we had finalized our manuscript. We hope you find the discussion satisfactory.

*3) What is known about taste receptors in fruit flies, and specifically, which of them are expressed in fruit fly larvae? C elegans expresses lots of different receptors for stimuli of the same valence (good/bad) on the same neuron. Therefore, if one were to do the same experiments on C elegans, that they would not be capable of encoding type of reinforcement, only the value. This would be an important prediction from the authors work*.

We agree that this prediction should be spelled out within our paper. What appears safe to say is that our data predict that the sensory neurons mediating fructose and aspartic acid reinforcement, respectively, cannot be completely overlapping, and that the sensory neurons mediating quinine and high salt punishment likewise cannot. We also added “taste” as keyword. We would like to refrain from a more detailed discussion, however. This is chiefly because taste coding is not the central topic of the paper.

Furthermore, the situation is complicated by the fact that, at least regarding bitter, the sensory neurons mediating reinforcement on the one hand versus those mediating innate avoidance on the other hand are distinct (ablating Gr33a‐Gal4 expressing neurons abolishes innate avoidance, [17] Chem Sens, but leaves bitter punishment intact, [1], Front Behav Neurosci).

Arguably, a similar situation can be expected for the other three tastants (as has recently been shown for water‐taste, too: [19]). We would thus need to reckon with a total of up to 8 ascending taste processing channels. To complicate matters even further, it remains to be seen whether the sensory neurons that are responsible for mediating the information about the gustatory environment during the test are the same as those mediating reinforcement, or the same as those mediating innate choice behavior – or yet different.

Indeed, the number of gustatory sensory neurons in the larva is about three times higher than the number of olfactory sensory neurons... Lastly, despite impressive recent advances, the mapping of gustatory receptors onto the gustatory sensory neurons of the larva is as yet incomplete (in the case of amino acids, practically nothing is known) and to some extent controversial (e.g. regarding sugar sensors). We thus hope that the manuscript as it now stands is clear in the way it states the most immediate implications of the present data for sensory physiology.

*4) Fruit flies, like other insects, have a lot of different taste receptors. They presumably need them to discriminate among many different tastants. So, in some sense, one would predict a priori that insects can discriminate value. Why would the receptors that enabled the 'type' discrimination evolve only to have them summed in a 'good' and in a 'bad' center in the brain for association? It might suggest an interesting experiment for future work that reflects on the need to discriminate type. There is a large literature on nutrition of carbohydrate, amino acids, etc., in insects. A Google search for authors Simpson or Raubenheimer and 'nutritional rails' will find relevant papers. Insects need to maintain a balance of these nutrients in their diet. If one experimentally drives insects off of their preferred nutritional balance (ratio of carbohydrate to amino acid, for example), insects will typically change their diet choices to bring them back into nutritional balance. If the authors were to deprive larvae of sugar or amino acids in their diet, then one might expect interference where there is currently none (e.g.*
Figure 1
*the 3rd and 6th experimental groups). Having shown that Drosophila larvae can encode type, which the authors conclusively did, then showing that insects can ignore the learned preferences for a deprived type would show even more complexity. But this is not an experiment that would be needed for a revision*.

Thank you for sharing this idea! We had something like this in mind when we were referring to “situationally‐defined matters of concern”, yet we realize this was phrased too densely, and not explicitly enough. Together with now pointing out the Simpson‐Raubenheimer literature, which indeed is fascinating, the way the paper reads now is hopefully a bit clearer.

(Regarding the rationale for why nature may indeed be merging sensorially different inputs towards a common‐currency signal, we would like to point out the paper by [18]. Such a common‐currency signal may be useful when facing the choice between outcomes that differ in kind. Secondly, it could be the case that the distinct sensors are used ‘merely’ for the immediate, reflexive behavior towards tastants, and that these signals then are summed for their reinforcing effects.)